# Osteogenic Induction with Silicon Hydroxyapatite Using Modified Autologous Adipose Tissue-Derived Stromal Vascular Fraction: In Vitro and Qualitative Histomorphometric Analysis

**DOI:** 10.3390/ma15051826

**Published:** 2022-02-28

**Authors:** Muhammad Marghoob Khan, Shadab Ahmed Butt, Aqif Anwar Chaudhry, Amir Rashid, Kashif Ijaz, Asifa Majeed, Hashmat Gul

**Affiliations:** 1Army Medical College, National University of Medical Sciences, Islamabad 46000, Pakistan; drmarghoobkhan@gmail.com (M.M.K.); drshadabbutt@gmail.com (S.A.B.); dramiramc@yahoo.com (A.R.); asifa_pak@yahoo.com (A.M.); dr.hashmatgul@gmail.com (H.G.); 2Interdisciplinary Research Centre in Biomedical Materials, Lahore Campus, COMSATS University Islamabad, Lahore 54000, Pakistan; kashifijaz@cuilahore.edu.pk

**Keywords:** tissue engineering, hemolysis, silicon hydroxyapatite, biocompatibility, cytotoxicity, histomorphometric analysis, stromal vascular fraction

## Abstract

Large bone defects requiring invasive surgical procedures have long been a problem for orthopedic surgeons. Despite the use of autologous bone grafting, satisfactory results are often not achieved due to associated limitations. Biomaterials are viable alternatives and have lately been used in association with Stromal Vascular Fraction (SVF), stem cells, and signaling factors for bone tissue engineering (BTE). The objective of the current study was to assess the biocompatibility of Silicon Hydroxyapatite (Si-HA) and to improve osteogenic potential by using autologous adipose-derived SVF with Si-HA in a rabbit bone defect model. Si-HA granules synthesized using a wet precipitation method were used. They were characterized using scanning electron microscopy (SEM), Fourier transform infrared (FTIR), and X-ray diffraction (XRD). A hemolysis assay was used to assess the hemolytic effects of Si-HA, while cell viability was assessed through Alamar Blue assay using MC3T3 mouse osteoblasts. The osteogenic potential of Si-HA both alone and with enzymatically/non-enzymatically-derived SVF (modified) was performed by implantation in a rabbit tibia model followed by histomorphometric analysis and SEM of dissected bone after six weeks. The results showed that Si-HA granules were microporous and phase pure and that the addition of Silicon did not influence Si-HA phase composition. Si-HA granules were found to be non-hemolytic on the hemolysis assay and non-toxic to MC3T3 mouse osteoblasts on the Alamar Blue assay. Six weeks following implantation Si-HA showed high biocompatibility, with increased bone formation in all groups compared to control. Histologically more mature bone was formed in the Si-HA implanted along with non-enzymatically-derived modified SVF. Bone formation was observed on and around Si-HA, reflecting osseointegration. In conclusion, Si-HA is osteoconductive and promotes osteogenesis, and its use with SVF enhances osteogenesis.

## 1. Introduction

For orthopedic surgeons, management of critical bone defects is a clinical challenge. Often, invasive surgical procedures are employed to reconstruct integrity of the bone structurally; however, satisfactory results remain difficult to achieve. Bone grafts are another option; although viewed as a “gold standard” treatment, these are associated with various limitations and complications. In autologous bone grafting the operating time is considerably longer, and limited quantities are available for grafting. Additionally, such procedures are usually associated with morbidity [1,2]. Bone Tissue Engineering therefore presents solutions to the aforementioned clinical predicaments. The presence of host cells, a signal mechanism to trigger differentiation of cells to bone forming cells, an artificial scaffold or matrix on which the new bone can form, and adequate blood supply are necessary for bone healing and/or bone graft incorporation [1,3]. In this context, “the diamond concept” is based on four biological essential components, namely, bone forming cells, an osteoinducive stimulus, osteoconductive material, and a hospitable environment [4].

Bone consists of cortical and cancellous bone. Cortical bone is highly dense and is organized into osteons, lamellae, and the collagen-mineral composite. The organic component of bone consists of collagenous and non-collagenous proteins along with bone matrix and cells. About 99% of the total body calcium is present in bones, with most of the mineral content in the form of biological apatite, which is similar to synthetic hydroxyapatite (HA), (Ca_10_(PO_4_)_6_(OH)_2_) [5]. Biological apatite, however, contains additional ions such as CO_3_^2−^, SiO_4_^4−^, and Mg^2+^, and the presence of these ions is linked to specific biological functions [1,6].

The limitations posed by bone grafts have led to the development of HA. Being similar in composition to bone mineral component, it is currently being widely used as a bone-substituting material. HA is known to play an instrumental role in new bone tissue formation, mineral deposition, and vascularization, leading to its potential use in the coating of implants, as granules or scaffolds for filling bone defects, and in synthetic bone grafts [7,8]. Derived from either natural or synthetic sources, HA can be synthesized through the precipitation method, hydro-solvo-thermal processing, solid state synthesis, sol gel techniques, and self-propagating thermal synthesis [9].

HA has nevertheless been observed to be less bioactive in relation to bone. This has led to HA being substituted with various ions such as Si^4+^, Mg^2+^, Sr^2+^ and Zn^2+^, which have a great impact on bone turnover and osteogenesis through their regulation of osteoblasts and osteoclasts and differentiation of MSCs. In this context, Si^2+^ has shown promising results with respect to promoting new bone formation during in vitro and in vivo tests, mimicking certain structural and compositional properties of bone [6,10,11]. Additionally, in order to determine the osseointegration capacity and osteoconductivity most studies have utilized porous biomaterials, usually in the form of scaffolds [12,13]. Hydroxyapatite or calcium phosphate-based products are available in a variety of shapes and forms. In this context, commercially available HA such as Actifuse, Apapore (ApaTech, Leics, UK), and Apacerum (Pentax, Tokyo, Japan), etc., have previously become available on the market. It is understood that orthopaedic surgeons require a variety of shapes and forms of bone regenerative materials when dealing with real-life surgical cases.

Age-related pathologies such as reduced bone mass through osteoporosis or loss of cartilage along with arthritic conditions impact the quality of life of the elderly population. In order to counter pathologies experienced in old age and those associated with loss of tissue, researchers are paying significant attention to stem cell-based regenerative treatment. Mesenchymal stem cells (MSCs) found in the bone marrow, adipose tissue, tendons, and ligaments have the differentiating capacity for osteogenic and chondrogenic lineages, becoming osteoblasts and chondroblasts [14,15]. Recently, research has been performed on adipose tissue-derived stem cells (ADSCs) due to certain advantages over other sources of MSCs. Adipose tissue has a variable percentage of SVF, and by utilizing in vitro cultures of SVF, ADSCs can be isolated. SVF isolation is an intermediate step for isolation and further expansion of ADSCs, as 100–1000-fold more MSCs can be isolated from adipose tissue as compared to bone marrow, and fat harvesting is easy. Additionally, as compared to adipose tissue, bone marrow-derived MSC procedures are more invasive and painful. SVF can be isolated from small volumes of adipose tissue either manually or via automated devices. In addition, commercially available protocols can be utilized. It has long been known that SVF provides an environment where ADSCs live and where their stemness is maintained [16,17].

We hypothesize that Si-HA granules are biocompatible, have no adverse biological effects, and, when used with lipoaspirate-derived SVF in order to enhance osteogenesis, present a superior treatment method for both the surgeon and the patient.

This study proposes an advanced solution for management of critical bone defects using an experimental rabbit tibia model without interference of any fixation device. We used Si-HA granules alone and along with autologous adipose tissue-derived SVF, respectively. A traditional method of SVF extraction utilizing collagenase was employed, and a modified non-enzymatic derived SVF was used along with Si-HA to achieve enhanced osteogenesis when used with Si-HA granules. The aim of the current study is to evaluate Si-HA for its biocompatibility, to assess its ability to regenerate bone, and to determine the effects of autologous adipose tissue-derived SVF in enhancing osteogenesis when used with Si-HA in critical-sized bone defects. We hypothesize that the Si-HA used in the current study is biocompatible, has no adverse biological effects, and is suitable for bone tissue engineering purposes, as well as, lipoaspirate-derived SVF enhances osteogenesis when used along with Si-HA. All these treatment options should, therefore, present the surgeon with a spectrum of surgical use choices based on the same material (i.e., Si-HA granules) in order to best determine how the granules are to be used for bone defects during surgery.

## 2. Material and Methods

The study was carried out at the Department of Anatomy, Army Medical College, Rawalpindi, in reciprocity with the Interdisciplinary Research Centre in Biomedical Materials (IRCBM), COMSATS University Islamabad, Lahore Campus (CUI Lahore Campus), Lahore and the National Institute of Health (NIH), Islamabad. It was a lab-based randomized control trial using the sampling technique of non-probability convenience sampling and with a study duration of two years, six months.

### 2.1. Synthesis of Silocon-Substituted Hydroxyapatite

Silicon-substituted Hydroxyapatite (Si-HA) was provided by IRCBM, CUI Lahore Campus. The common wet precipitation method was used for synthesis, using Calcium hydroxide, Phosphoric acid, and Tetra Ethyl Ortho Silicate (TEOS) as calcium, phosphorus and silicon precursors, respectively, with the precipitation reaction being carried out at room temperature. After filtering with a vacuum filtration assembly, the synthesized samples were oven-dried and heat-treated at 1100 °C to form granules prior to use [18,19].

### 2.2. Material Characterization

#### 2.2.1. XRD

XRD patterns were obtained utilizing Philips PANalytical X’Pert Powder system in order to assess the phase purity of the synthesized bioceramic material. Using CuKa radiation with operating conditions 30 kV and 22 mA, XRD spectra were collected over 2 θ, ranging between 5–70 at a step size of 0.02 with a count time of 2 s/step.

#### 2.2.2. SEM

Si-HA granules were investigated for their size, morphology, and porosity using scanning electron microscopy. The granules were gold coated (250 A°) in an Ion sputtering device (Quorum SC 7620, Sussex, UK) and analyzed with an SEM (TESCAN, Vega 3 LMU, Brno, Czech Republic) operating at 7 kV. Imaging was done at magnifications of 1 k×, 5 k×, 10 k×, and 25 k×.

#### 2.2.3. FTIR

The FTIR technique was utilized using a Thermo Nicolet 6700 with ATR accessory to analyze the functional groups in synthesized samples in the as-precipitated state and the heat treatment state for confirmation of the effects of the silicon. FTIR spectra were collected in the 4000–500 cm^−1^ range, averaging 128 scans with a resolution of 4 cm^−1^.

### 2.3. In Vitro Studies

#### 2.3.1. Hemolysis Essay

A hemolysis test was carried out to determine the in vitro hemocompatibility of Si-HA; 5 mL of blood from a healthy donor free of infectious diseases (HIV, HCV, HBV) and other blood-related pathologies was centrifuged (EBA 20, Hettich Zentrifugen) for 4 min at 5000 rpm. Then, 5 mL of phosphate buffer saline (PBS) solution was added to the obtained RBC pellet and centrifuged at 6 min at 5000 rpm at 4 °C; 0.1 mg of Si-HA was then incubated with 0.2 mL of prepared blood sample and 0.8 mL of PBS for 1 h at 37 °C (JSON 100, JSR). PBS solution and deionized water were used as negative and positive controls, respectively. After 1 h incubation, centrifugation at 5000 rpm for 6 min was carried out on all of the samples. Optical density (OD) was measured at 570 nm using UV spectrophotometry (UV Mini 1240, Shimadzu, Japan) on the supernatant solution. The Hemolysis degree of Si-HA was determined by applying the following formula as per the study of Bandgar et al. [20]:(1)$$\%\mathrm{Hemolysis}\;=\;\frac{\left(\mathrm{Sample O.D})\;-\;(\mathrm{Negative control O.D}\right)}{\left(\mathrm{Positive control O.D})\;-\;(\mathrm{Negative control O.D}\right)}\;\times \;100$$


#### 2.3.2. Blood Complete Picture

To further validate the effects of Si-HA on red blood cells, a complete blood picture was obtained by implanting the said biomaterial in rabbit tibia and obtaining blood samples from the marginal vein of ear, before and six weeks after the procedure.

### 2.4. Cytotoxicity Analysis

An Alamar Blue assay (Thermofisher, No. 5501959) was performed for quantitative determination of cell viability using exponentially-growing MC3T3 cells.

#### 2.4.1. Cell Culture

MC3T3-E1 mouse pre-osteoblast cell lines (American type culture collection ATCC # CRL-2593TM) procured from IRCBM, CUI Lahore, were cultured with low glucose minimum essential medium (α MEM) (Gibco, NY, USA) supplemented with 10% fetal bovine serum (FBS) and 1% penicillin/streptomycin (Gibco BRL). The culture was carried out at 37 °C in a humidified 5% CO_2_ incubator, with the culture media replaced every 24 h. The cells were digested with 0.05% trypsin/0.53 mM EDTA and the fifth passaged cells were used for the assay [21].

#### 2.4.2. Alamar Blue Assay

Before placing Si-HA in a 24-well plate, it was sterilized with UV light. Estimation and counting of cells were performed using trypan blue staining and hemocytometer, respectively. After seeding the cells in triplicate at a density of 2 × 10^4^ cells per well, they were incubated at 37 °C in humidified 5% CO^2^. Alamar blue solution prepared as per the manufacturer’s protocol (Thermofisher, No. 5501959) was added at a concentration of 200 µL/2 mL to the individual wells on the second and eighth day. After incubating the samples for 5 h, absorbance at 550 nm and 620 nm was measured using a Biorad PR4100 absorbance microplate reader. Cells without Si-HA were used for a positive control, while culture medium without cells was used for a negative control [22].

### 2.5. In Vivo Studies

All the animals were treated in conformity with the policies and principles of laboratory animal care and animal research guidelines (Reporting In Vivo Experiments, ARRIVE Guidelines), and the study was conducted after approval from the Ethical Review Committee of the Army Medical College, Rawalpindi (ERC, AMC). Twenty female New Zealand white rabbits (about ten months old, average weight 1600 to 2500 g) were used and kept at National Institute of Health animal house under a natural light dark cycle of 12 h at about 22 °C. The rabbits were fed with standard laboratory feed and tap water, and were continuously monitored by a veterinarian during the experimental period. Before the surgical procedure all rabbits were quarantined for seven days for acclimatization and to ensure that the rabbits were free from any disease [23,24,25].

#### 2.5.1. Sample Size Calculation

G-power version 3.1.9.2 was used for calculating the sample size. For one-way Anova, considering the values of α as 0.05, power of test as 0.8, and effect size as 0.9 for four groups, five rabbits per group was calculated, meeting a total sample size of 20 (Table 1) [26]. The details regarding sample grouping are given in Table 1.

#### 2.5.2. Surgical Procedure: Fat Harvesting

Adipose tissue was isolated from ten adult female New Zealand rabbits from Groups C and D (Table 1). After sedating the rabbits with intramuscular injection of Xylacine 5 mg/kg (Xylax^®^ 2%, 25 mL) and Ketamine 35 mg/kg (Ketolar^®^ 50 mg/mL), an incision was made in the right inguinal region. The inguinal fat pad was identified and about 2 to 3 cm^3^ of fat was removed. The isolated fat was washed thoroughly five times with PBS and transported back to the laboratory for fat processing. Wounds were cleaned and closed with 2/0 silk [27].

### 2.6. SVF Isolation: Enzymatic Digestion

Adipose tissue 2 gm from the rabbits of Group C was washed in PBS containing 5% penicillin/streptomycin (Gibco BRL) (*v*/*v*), finely minced, and digested for 40 min using collagenase type I (0.075% *w*/*v*) (Sigma, St Louis, MO, USA) at 37 °C (JSSI-100C, JSR shaking incubator). The control medium was made up of Dulbecco’s modified Eagle’s medium (DMEM; Gibco BRL, Gaithersburg, MD, USA), 10% FBS (Gibco BRL), and 1% P/S (*v*/*v*), and was used in equal volume to neutralize collagenase. After centrifuging at 1200 rpm for 10 min (EBA 20, Hettich Zentrifugen), the obtained high-density SVF pellets were suspended in red blood cell lysis buffer (ELB), centrifuged again, and filtered with 100 μm nylon mesh. The pellets were incubated in control medium in petri dishes along with bioceramic and transported to the animal house for reimplantation in the respective rabbits [26,28,29].

### 2.7. SVF Isolation: Non Enzymatic (Modified)

Adipose tissue 2 gm from the rabbits of group D was washed and minced as before and transferred to a 50 mL tube. ELB (vol/vol) was added for 10 min, and after removal of ELB the sample was placed in 4 mL of control medium to be centrifuged at 2000 rpm for 5 min (EBA 20, Hettich Zentrifugen). The tube was shaken thoroughly and the sample was centrifuged again and allowed to stand for 5 min. Floating debris was removed, and after resuspending the pellet in 4 mL control medium it was centrifuged again. In order to isolate the maximum concentration of cells along with residual adipocytes and matrix components, the SVF pellets were modified by not filtering with 100 μm. The pellets were collected and resuspended in control medium containing Si-HA for reimplantation in the respective rabbits [15,30].

### 2.8. Surgical Procedure for Bone Defects

After placing each rabbit in a lateral position, the right leg of each rabbit was disinfected with 10% povidone–iodine. The rabbits were anaesthetized by intramuscular injection of Xylacine 5 mg/kg (Xylax^®^ 2%, 25 mL) and Ketamine 35 mg/kg (Ketolar^®^ 50 mg/mL). After identifying the right tibial tuberosity, a 5 cm vertical skin incision was made with a #15 scalpel blade along the length of the medial side of tibia. After cutting the fascia and retracting the tibialis anterior muscle, a critical-sized defect of about 9 mm length, 6 mm breadth, and 4 mm depth was randomly performed in the proximal metaphyseal–diaphyseal area using a surgical drill on low speed (Escort-III Micromotor, Saeyang Marathon, H20, South Korea) along with copious irrigation with normal saline to prevent thermal bone necrosis. The surgical site was carefully irrigated with normal saline to wash away any contamination and bone debris, after which the bone defects of each rabbit were filled with Si-HA along with SVF (Table 1) [24].

All wounds were closed using metallic staples, which were removed ten days post-operatively. All animals were housed individually in pans with no postoperative restriction on activity and were fed ad libitum. Injections of Tramadol (5 mg/kg) for pain control and Enrofloxacin 10 mg/kg (Baytril^®^ 5%, Germany) as a prophylactic antibiotic were given intramuscularly daily up to seven-days post-surgery [25,26]. Si-HA was sterilized using gamma irradiation prior to implantation in the bony defects.

### 2.9. Animal Sacrifice and Sample Manipulation

Three rabbits died immediately after surgery, two due to anesthesia and one due to a reaction to the pain killer; these were replaced in order to meet the required sample size. All rabbits were sacrificed after six weeks by Chloroform inhalation in a closed chamber, following which subperiosteal dissection of the tibias was carried out. With the defect area implanted with Si-HA as the center, the dissected tibias were cut into two equal half-sections. One section from each sample was preserved for SEM analysis in neutral buffered formalin, while the other was decalcified with 10% nitric acid for four days and processed with a graded series of ethanol and xylene (70 to 100%) and embedded in paraffin wax for light microscopy [26,27].

### 2.10. Histomorphometric Analysis

For morphological analysis and cell counting, five consecutive 5-µm sections were obtained from each specimen along the longitudinal plane, out of which two sections were stained with Hematoxylin and Eosin (H&E) and one section was stained using Masson’s Trichrome Blue (Sigma) for histologic evaluation. Alternatively, for tartrate-resistant acid phosphatase (TRAP) activity, two sections from each specimen were histochemically stained with TRAP stain (Sigma Aldrich TRAP Kit—387A-1KT) and counterstained with Gills hematoxylin per manufacturer’s instructions [31,32].

For semi-quantitative variables such as the presence or absence of new vessels, granulation tissue, and osteoid tissue, all slides were first observed with 10× magnification low power field, then readings were recorded at 40× magnification and 100× (oil immersion) using a light microscope (Olympus, Tokyo, Japan) [33]. Each variable in the bone defect area was defined as follows: neovascularization as capillaries and medium-caliber vessels lined by endothelium; inflammatory infiltrate as inflammatory cells (lymphocytes and monocytes); and bone formation as bone tissue. Osteoblasts on the surface of the trabeculae were considered active if they were polygonal in appearance and inactive if they had a flattened appearance; osteocytes were defined as inactive cells in the bone lacunae and osteoclasts as multinucleated giant cells in gaps between the newly-formed bone or on trabeculae surfaces [34].

A modified histological evaluation by Lucaciu et al. (Table 2) with ten parameters and a maximum healing score of 37 was used for assessment of bone regeneration [35]. The score given to each parameter was added, and a histological healing score for each rabbit was thus obtained.

Additionally, in the histological specimens inflammatory reactions, granulation tissue, new bone formation, and any abnormality such as enlarged Haversian canals (if present) were assessed with a modified histological scoring system by H. Ch. Vogely et al. (Table 2) [36]. The scoring system includes the presence or absence of granulation tissue, lymphocytes and monocytes, micro-abscesses, and fibrosis in the medullary canal and cortex, as evidence of infection, if any. The severity index was Score 0 (absence of infection/inflammation) to Score 3 (severe infection/inflammatory reaction) (Table 3). Stained sections were analyzed and photographed with a microscope (Olympus-CX41, Olympus, Japan) connected to a CCD camera (DP72; Olympus, Japan) using cell Sens standard software (Olympus, Japan) [33].

### 2.11. SEM of Bone

To analyze the relationship between Si-HA and bone tissue, non-decalcified tissue was examined in SEM. After fixation of samples with 2.5% glutaraldehyde in PBS for 48 h, the samples were washed with buffer twice before dehydrating with acetone. Critical-point drying of the sample was carried out with solvent-substituted liquid Carbon Dioxide (Blazer, FI 9496, Liechtenstein). Afterwards, samples were coated with platinum (LTD, VE 1010, Japan) and observed with an SEM (Hitachi, 2380N) operating from 1.8 kv to 7.0 kv at varying magnification and resolution [33].

### 2.12. Statistical Analysis

Analysis of the data was carried out by applying ANOVA and post hoc Tukey’s test to evaluate any statistically significant differences within and between the groups, with a *p*-value < 0.05 being taken as significant; correlation tests were performed as well. Statistical analysis of independent variables was carried out using Statistical Package for the Social Sciences (SPSS) version 25, Statistica 12.0, and Microsoft Excel software.

## 3. Results and Discussion

Autologous bone grafts, despite being used in cases with bone tissue loss, have certain limitations such as limited availability, added morbidity, etc. Synthetic bone grafts offer a useful alternative. Silicon-substituted hydroxyapatite (Si-HA) has osteoconductive properties, excellent biocompatibility, and is similar in mineral composition to bony tissue [8]. In this work, we explore the potential use of Si-HA, both as a standalone solution and with adipose tissue/SVF for defect filling. We hypothesized that the Si-HA used in the current study would be biocompatible, have no adverse biological effects, and be suitable for bone tissue engineering purposes, and that lipoaspirate-derived SVF would lend additional osteogenic potential to the treatment when used along with Si-HA.

### 3.1. XRD

X-ray diffraction (XRD) was used to assess the phase purity of the Si-HA. The XRD patterns of as-precipitated samples (Si-HA) and heat-treated samples (Si-HA—HT) are given in Figure 1; it can be seen that there is a good match with phase-pure hydroxyapatite when compared to ICDD pattern #09-432 for phase-pure HA.

After heating at 1100 °C temperature, single-phase Si-HA was obtained with no evident secondary phases such as tricalcium phosphate (TCP) or calcium oxide (CaO); the precipitated Si-HA exhibited broad diffraction lines indicative of small particle size and/or low crystallinity. The XRD peaks for heat-treated Si-HA samples were sharper than the corresponding peaks of pure Si-HA, suggesting increased crystallinity and growth in particle size [37,38].

### 3.2. FTIR

Figure 2 shows the collected FTIR spectra.

A broad absorption peak centered at 3337 cm^−1^ associated with the O-H group was observed. This broad peak vanished for the heat-treated sample, potentially due to evaporation and/or loss of lattice water. Peaks at 1088, 1037 and 961 cm^−1^ were attributed to the stretching of P-O bonds in phosphate groups of Si-HA. A shoulder observed at 935 cm^−1^ is possibly due to silicate groups replacing phosphate groups and an associated effect on the crystal structure. Lastly, the 1435 cm^−1^ peak can be attributed to substituted carbonate ions in the Si-HA crystal lattice. This peak is absent in the heat-treated sample, suggesting loss of CO_2_ after heat treatment. From these results, we are able to conclude that our material is indeed apatitic in nature in its as-precipitated form (as observed using X-ray diffraction), and that silicon substitution does take place [38,39].

### 3.3. Scanning Electron Microscopy

The Si-HA granules were sieved (500 µm to 1 mm) prior to surgical use. Figure 3 shows the surface of a granule at varying magnifications. This reveals that the Si-HA granules contained micron/sub-micron-level porosity.

One of the most important factors of these biomaterials is that they match the native tissue characteristics and allow osteoinduction, for which the microarchitecture of the biomaterials such as their pore size and porosity is crucial [40,41]. It is believed that the porosity and microporosity of biomaterial particles enhances osteogenesis by enhancing ionic exchange with body fluids and increases vascularization [38,39].

It is evident from the formation of vessels and new bone that our Si-HA granules facilitated growth of vessels and exchange of nutrients, possibly due to the presence of porosity in the granules (as seen in the histological data provided below) [39,42,43]. Additionally, the similar composition of Si-HA and bony tissue leads to new bone formation in the bony defects. This is achieved through the enhancement of calcium and phosphate ion deposition along with the remodeling action of host osteoclasts and osteoblasts [43,44,45]. Indeed, a histomorphometric and micro-CT analysis by Chackarti et al. showed that the same pattern of bone formation was produced with granules of different sizes (small or large), as a “bone bridge” network along the granules was produced by the bone surrounding the graft [38,43].

### 3.4. Cytotoxicity Analysis: Alamar Blue Assay

Figure 4 displays the Alamar Blue fluorescence measured on the second and eighth day. The cell viability of MC3T3 cells on day two was 94%, slightly less than the control. On the other hand, the cell viability detected through fluorescence intensity was significantly higher on day eight compared to day two. The result of the Alamar Blue assay confirms the biocompatibility of Si-HA, as no toxic effects were observed.

Different tests are available to determine cell viability. Several of these tests damage cells during procedures or as per requirements, limiting their application. This has resulted in the introduction of non-toxic dyes which allow analysis of cells over time while keeping the cells in culture. Alamar Blue is one such vital dye, which due to the metabolic activity of living cells is reduced to fluorescent pink. We used Alamar Blue in our studies because it is non-toxic to other cells compared to dyes used in other cell viability assays such as the MTT test. Additionally, dyes such as tetrazolium salts are not reduced by cytochromes, while Alamar Blue can be reduced by cytochromes, resulting in high sensitivity to lower concentrations of cells [46]. In our study, the growth of cells was slower at day two compared to control (Figure 3). This slow growth was due to the fact that MC3T3 cells were adapting to the new microenvironment. The biocompatibility of Si-HA was evident by the exponential growth of cells at day eight, confirming that the cells were both metabolically active and proliferative. Our findings concur with other reported results [47]. In summary, the presence of Si-HA had no ill effects towards cell viability and growth, as the concentration of cells increased with the passage of time [22,48].

### 3.5. Hemolysis Essay

The results obtained through the hemolysis assay are shown in Table 4.

An essential characteristic of materials intended to be implanted or in direct contact with blood is their hemocompatibility. A hemolysis essay shows the degree to which red blood cells are lysed when in contact with a sample; therefore, evaluation of the hemolytic potential of any material to be administered in this way is required. The hemocompatibility of Si-HA was evaluated to determine any different effects on hemolytic activity and RBC membrane integrity [42]. The results are expressed as the hemolytic ratio, labelled in three categories, with <2% indicating non-hemolytic, 2–5% slightly non-hemolytic, and >5% hemolytic. In our study, the observations were in agreement with previously reported results [41,49,50]. A maximum hemolysis value of 3.66% in Sample 2 showed increased thrombogenicity, which may be the result of greater blood–surface interactions due to the ionic groups in Si-HA interacting with certain blood components (Table 4) [51]. Complete blood pictures of all the experimental groups carried out before and after the placement of Si-HA showed no differences in the blood pictures after six weeks, confirming the non-hemolytic effect of Si-HA (please refer to Appendix A). Overall, the hemolysis percentages of all samples were less than 5%, which along with the normal blood pictures indicates that the prepared Si-HA does not have a hemolytic effect on red blood cells and, is thus hemocompatible.

### 3.6. Histomorphometry

#### 3.6.1. H&E Staining

Increased new bone formation (% bone) was evident in the histomorphometrical analysis of Groups II, III, and IV when compared to group I (*p* < 0.05) (Table 5 and Table 6). Bone formation remained minimal in the control group, being not filled with Si-HA (Figure 5a). After six weeks, remaining particles of Si-HA and residual spaces of varying shapes and sizes with surrounding laminar bone were noted in the defect area of Groups II, III, and IV, indicating that Si-HA as a bone substitute is osteoconductive (Figure 6a,b,d,e). In all experimental groups, although a few scattered inflammatory cells were present, proper bone healing with no inflammatory signs or foreign body reaction was observed, suggesting biocompatibility of the material (Figure 5c). Blood vessels with revascularization foci along with mature Haversian systems and surfaces covered by osteoblasts, osteoclasts, and osteocytes in varying proportions were present in all experimental groups, implying active osteogenesis.

In histological evaluation grading, no bone formation or osteocytes, osteoblasts, or osteoclasts were observed centrally, while the same were detected only at the periphery in all rabbits of the control group (mean grading 10) (Table 5). At higher magnifications (40× and 100× oil immersion), no evidence of inflammation or granulation tissue was observed in any of the subjects except for a few scattered inflammatory cells (Figure 5c). In histological evaluation grading, statistically significant differences were observed between all experimental groups (*p* value < 0.001) and the control, while no differences were observed in the intergroup comparison (Table 5 and Table 6).

In the histological evaluation of sections (severity index) in all groups, a normal cortex with small Haversian canals was observed, with no evidence of periosteal reaction (Figure 5f). In our study, the minimum grading obtained was 1.2 and the maximum was 2.4 (Table 7). Although a few leukocytes were occasionally found in the medullary canal, there was no evidence of microabscesses, granulation tissue, fibrosis, or inflammatory exudate. Minimal formation of new bone at the periphery of the bone defect with normal architecture was observed in the control group at six weeks post-surgery (Figure 5a). Additionally, in all the experimental groups most of the new bone tissue formed was mature, containing abundant osteocytes in lacunae (Figure 5f) and with bone marrow cells having good cellular concentration, as evidenced by their basophilic coloration (Figure 6d,e,f). The above findings are a good indication that the implanted material did not incite any foreign body reaction and that biomaterial sterilization was effective throughout the allocated time period. The results of the histological semi-quantitative scoring are summarized in Table 7.

#### 3.6.2. Masson’s Trichome Staining

Our results showed the deposition of collagen fibers and mature osteoid tissue via Masson’s Trichome stain at the defect centers of group II, III and IV in accordance with the H&E staining (Figure 5, Figure 6 and Figure 7). Mature bone with mostly dark red and blue coloration was observed in the control group at the periphery. Zhang et al. reported a similar pattern of collagen and mineral deposition in their study, which is in accordance with our results [52].

#### 3.6.3. TRAP Staining and Osteoclast Count

We observed all types of bone cells in varying numbers in all groups. Osteoclasts were observed in TRAP staining (Figure 8a,b) and H&E staining (Figure 8c,d), as indicated by the presence of Hawships lacunae and a multinucleated appearance. We observed more osteoclasts in group I and group II compared to group III and group VI. We expected an increase in osteoclast count in group III as well, for the same reasons as for increase in bone turnover. The low number of osteoclasts in the later groups may be due to the fact that more mature bone was present as well as, the number of osteoclasts is considerably lower than osteoblasts or osteocytes [53].

Histomorphometric analysis regarding bone formation showed a statistically significant difference between the experimental and control groups during the designated time period. Our results corroborate the findings of other researchers. To determine differences in bone formation between different HA formulations, A. Rahimnia et al., (2021) compared Synthesized HA (sHA), xenograft, and commercially available HA (cHA) in rabbit 8 mm calvarial defects. The authors reported that after four weeks, ossification in rabbits implanted with sHA was higher compared to other groups, while compared to xenograph implants the rate of new bone formation was higher in the cHA and sHA samples. Nevertheless, at twelve weeks the sHA group showed significantly higher new bone formation and ossification compared to the xenograft and cHA groups [54]. Similarly, Veremeev et al. (2020) compared commercially available bone autografts with purified bone collagen, DBM, and refined HA in 8 mm-diameter rat calvarial defect models and concluded that both HA and DBM are efficient in the repair of bone defects [55]. These results corroborate our findings, as increased bone formation was evident in all experimental groups. However, little or no data are available regarding comparison of Si-substituted HA and commercially-available synthesized HA.

In this context, Silicon-substituted biomaterials have been shown to increase osteoconduction and promote new bone formation in both in vivo and in vitro models. Platelet-rich fibrin membranes and silicon substituted ABG (PSB) showed increased mineralization and bone formation as compared to platelet-rich fibrin membrane and silicon (PB) and silicon substituted ABG (SB) when implanted in rabbit calvarial defects [56]. Similarly, in a study in a diabetic rat model, titanium implants with coatings of highly crystalline Si-substituted nHA (Si-nHA) improved osteogenic and angiogenic differentiation of BMSCs by Si-nHA in vitro, while an in vivo study demonstrated increased bone formation and bone–implant osseointegration compared to HA coatings [50].

The intensity of the inflammatory infiltrate at the site of implantation is another factor that must be considered when evaluating a biomaterial. Although few inflammatory cells were present, the absence of gross inflammatory reaction and granulation tissue in all experimental groups indicated that Si-HA granules, despite being a foreign body, do not induce an exacerbated inflammatory response, demonstrating the tissue biocompatibility and osteoconductive properties of Si-HA. In this context, our findings corroborate the findings of other studies [26]. Mature osteocytes are a type of bone cell present in the bone lacunae, where they are considered ‘entrapped’ osteoblasts. These are inactive and post-proliferative cells, and represent the most mature differentiation state of the osteoblast lineage. In humans, their numbers vary; however, it is estimated that there are roughly 25,000 osteocytes per mm^3^ of bone. Structurally, the appearance of osteocytes varies depending on their localization relative to the surface layer. Derived from osteoblasts, osteocytes which have been recently incorporated in the bone matrix have high organelles, while the more deeply-situated osteoblasts have a more flattened appearance [53,57]. In this context, we encountered both types of osteocytes in our study.

Neovascularization plays an important part in bone repair and when it is modest or absent, osteogenesis is either minimal or absent. Silicon-substituted scaffolds were found to be biocompatible, with enhanced vascularization compared to non-silicon substituted matrix in animal models [58]. The angiogenic and increased bone formation potential of SVF was studied by Farré-Guasch, (2018) in a human maxillary sinus floor elevation model, where increase bone percentages correlating with increased blood vessel formation were noted with a tricalcium phosphate scaffold along with SVF compared to control [59]. In all experimental groups, we observed foci of neovascularization and bone neoformation, reflective of neovascularization.

Our findings with regard to the use of SVF are in accordance with those of other researchers utilizing SVF for regeneration purposes. To determine the effects of SVF on osteogenesis, different biomaterials have been evaluated along with SVF. Rhee et al. reported new bone formation induced by SVF and DBM in an animal model [28]. E. Nyberg et al. (2019) utilized both SVF and passaged ASCs along with decellularized bone matrix and polycaprolactone scaffolds in murine critical-sized cranial defects as well as in in vitro studies. In vitro mineralization of the scaffold along with in vivo bone formation were observed with both SVF and passaged ASCs [60]. In another study, use of SVF on osteonecrosis jaw-like defects in mice increased the number of osteocytes, bone filling, bone mineral density, collagen fibers, and blood vessels observed after two and four weeks compared to the control [61]. The bone healing potential of SVF has been evaluated without the use of bioceramics. T. Kamenaga et al. (2021) transplanted human SVF (both fresh and cryopreserved after isolation) along with atelocollagen gel and PBS in rats with non-healing fractures; after week eight, they observed both radiologically and histologically greater fracture healing in the experimental groups compared to control. The authors’ findings suggest that for the treatment of nonunion fractures transplantation of SVF cells can be a viable option, and that cryopreserved SVF can be used without loss of cells [62].

Similarly, Ilaria Roato et al. studied the osteogenic potential of SVF and concluded that SVF is more osteoinductive then adipose stem cells plated with osteogenic medium [63]. In our study, we did not use any such stimuli other than use of growth media (DMEM) during isolation of SVF and transportation of SVF along with Si-HA; nevertheless, enhanced bone healing on the microscopic level was observed with SVF compared to bioceramic alone, which is in accordance with other studies utilizing SVF [27,29,63].

The regenerative ability of SVF depends on both ASCs and on released soluble factors such as the paracrine factors VEGF and ET-1, both of which can promote differentiation of osteoblasts. Other cells present in SVF include RBCs, leukocytes, MSCs, endothelial cells, and pericytes. It is due to these cells and soluble factors that SVF provides more benefits compared to ASCs. Cell culture and subsequent multiple passages are associated with certain risks and inconveniences, which are avoided with SVF [63,64]. Hence, ASCs and SVF should be considered as two separate entities, as both exhibit different features that influence their efficacy differently.

In our study, we modified the non-enzymatic isolation process of SVF in group IV via a centrifugation process in which the SVF pellets obtained in the end were not passed through a 100 µm mesh to ensure the maximum availability of cells in the pellet. Often, the end result of this process was isolation of SVF with intact cell–cell and cell–matrix communication. Conversely, due to utilization of enzymes, enzymatic isolation procedures disrupt all such communications, resulting in only single cells [65]. The higher percentage of new bone formation and the more mature Haversian system in group IV (followed by group III) signifies the osteogenic potential of SVF. This further signifies importance of cell–cell and cell–matrix interaction, which stimulates mesenchymal cells. Additionally, the presence of extracellular matrix in non-enzymatically derived SVF may act as a natural scaffold for the deliverance and guidance of cells, especially ASCs [66].

### 3.7. SEM: Biomaterial Interface

SEM of non-decalcified bone to observe the bone–biomaterial interface showed that Si-HA remained present at the implantation site six weeks post-implantation, with more or less the same morphological pattern in all three groups (Figure 9). New bone formation was present, with the bone surface clearly demarcated from the biomaterial (Figure 9a–d); the bony network grew on the granule surface (Figure 9d). Bone-like tissue was observed to adhere to the Si-HA granules in the vicinity and was closely attached to the Si-HA particles, implying newly-formed woven bone (Figure 9b,d). No apparent ultra-structural differences could be identified between the recently formed and pre-existing old bone. Additionally, there was absence of any inflammatory or fibrous connective tissue indicative of absence of bone tissue intolerance to Si-HA.

For the overall success of biomaterials in bone regeneration, osseointegration is considered a determining factor in bone response to implanted material. Understanding osteogenesis and the reaction of bone to implanted material may lead to development of new and improved biomaterials. Implant material success is strongly influenced by the relationship of implant material and tissue. In short, bone graft substitutes are dependent on the establishment of strong bonds with pre-existing bone. Although the in vivo performance of a biomaterial takes into consideration properties such as chemical composition, morphology, pore size, etc., the extent of osseointegration of the implanted biomaterial with the adjoining bony tissue is considered the standard measure of success [67]. The Si-HA used in the current study is a bioactive material, and the bone growth on the Si-HA granules and in the surrounding regions, as evident in the SEM analysis, shows both its osteoconductive nature and the precipitation of an apatite layer on its surface, forming a chemical bond. To date, the scientific community has not reached any agreement on the in vivo behavior of CaP-based biomaterials, particularly hydroxyapatite. Although the surface behavior of hydroxyapatite has been extensively studied, conflicting views on the interfacial apatite layer on the surface, its constituents, and its and orientation with respect to biomaterials still remain [51,67].

The demarcation between bone and degradable biomaterial migrates with the passage of time, advancing into the implanted material. This gives rise to a characteristic interlocking pattern between the new bone and the implanted biomaterial. This feature is common to many CaP-based biomaterials [68] and was observed in our study as well. In addition, significantly narrower fissure-like spaces were found to be filled with newly formed bone. These findings clearly indicate that the Si-HA used in the current study is both osteoconductive and capable of Osseointegration [67,68].

The evolving field of BTE has brought novel methods of manufacturing and modifications to the structure of biomaterials. Manufacturing technologies such as 3D printing have the added advantage of preparing composite scaffolds to match bone defects and bone porosity and structure, by taking into consideration 3D images and CT data analysis. Isolating a bone through tissue engineering is possible using bone bioreactors, which can provide a conducive medium for the integration of growth factors, cells, and scaffolds. Additionally, in orthopedics, genetic engineering and gene expression modification such as targeting genes that down-regulate bone formation are prospective alternate therapies [1].

It is pertinent to mention that although the literature review highlighted the use of Si-HA along with stem cells for in vitro analysis and the use of HA along with SVF for bone tissue formation [69,70], little or no data are available regarding use of Silicon-substituted HA along with SVF for BTE. The current study may be the first evaluation of Si-HA with SVF for bone regeneration. Additionally, compared to enzymatically-derived SVF, Si-HA construct, Si-HA alone, and control, our modified non-enzymatic-derived SVF Si-HA construct showed a greater degree of new bony tissue formation histologically. Not only did the Si-HA used in present study prove to be bioactive and osteoconductive with new bone tissue formation; this study demonstrated the feasibility of isolating modified non-enzymatic SVF along with traditional enzymatic SVF isolation as well, overcoming the disadvantages of cell culture. Our study showed that SVF therapy is feasible and effective, and that its simple methodology for isolation provides an on-table option for surgeons. Hence, Si-HA and SVF therapy could represent a more competent treatment approach and a proposed novel paradigm for dealing with critical-sized bone defects.

### 3.8. Conclusions

The bone regeneration study with Si-HA in a rabbit tibia model showed that Si-HA has superior osteoinductivity. Si-HA can both stimulate angiogenesis and enhance early bone formation compared to control. Although no statistical differences were found in bone healing between enzymatic and non-enzymatic-derived SVF, quantitative histological analysis showed more mature bone with increased new bone percentage and osteocytes in rabbits treated with modified non-enzymatic-derived SVF. In addition, Osteogenesis was evident in SEM analysis of bone biomaterial interface, highlighting the Osseointegration capacity of Si-HA.

These observations collectively suggest that using the novel technologies of bioceramic utilization either alone or alongside SVF can improve bone repair without requiring time-consuming osteogenic differentiation or costly growth factors. The methodology proposed herein, especially isolation of modified non-enzymatic-derived SVF, offers surgeons a critical comparative assessment of the use of synthetic bone grafts with and without adipose tissue/SVF, and through this, enables the assessment of multiple treatment options for patients.

### 3.9. Study Limitations

Although the initial data are reassuring in that surrounding orthotropic tissues and different cellular constituents and factors in SVF stimulated osteoinduction, the composition of the SVF cell population could have been influenced by SVF extraction steps such as centrifugal speed, time, and temperature, which were not determined in our study. Additionally, mechanical testing of the bone constructs for integrity evaluation was not carried out in this study.

### 3.10. Future Recommendations

Further studies looking into the molecular mechanisms behind healing induced by Si-HA (both with and without SVF) should provide a deeper understanding of the related healing mechanisms. As the developed bioceramics in this study have been shown to be biocompatible, further studies might aim at additional or different site delivery mechanisms such as injectability, syringability, etc.

## Figures and Tables

**Figure 1 materials-15-01826-f001:**
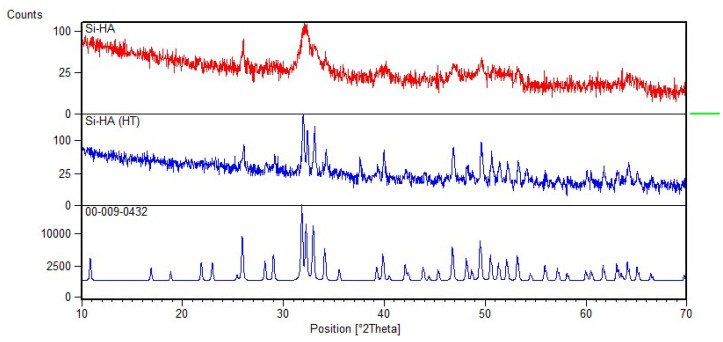
XRD graph of Silicon hydroxyapatite (top red), silicon hydroxyapatite, heat-treated sample (middle), and ICDD power diffraction reference (bottom).

**Figure 2 materials-15-01826-f002:**
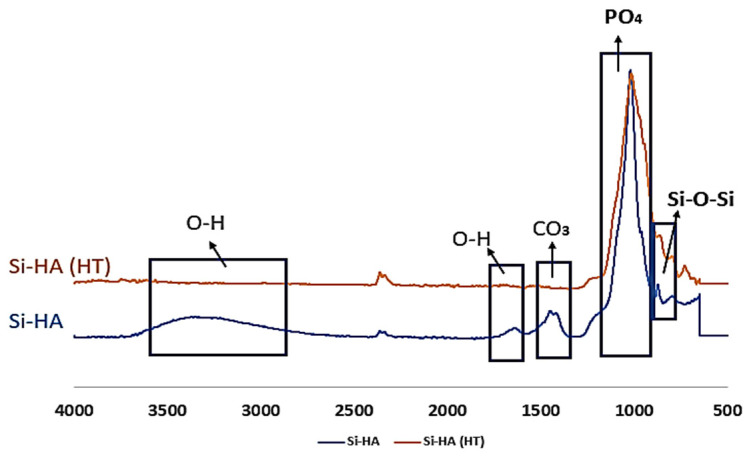
FTIR spectra of Si-HA granules.

**Figure 3 materials-15-01826-f003:**
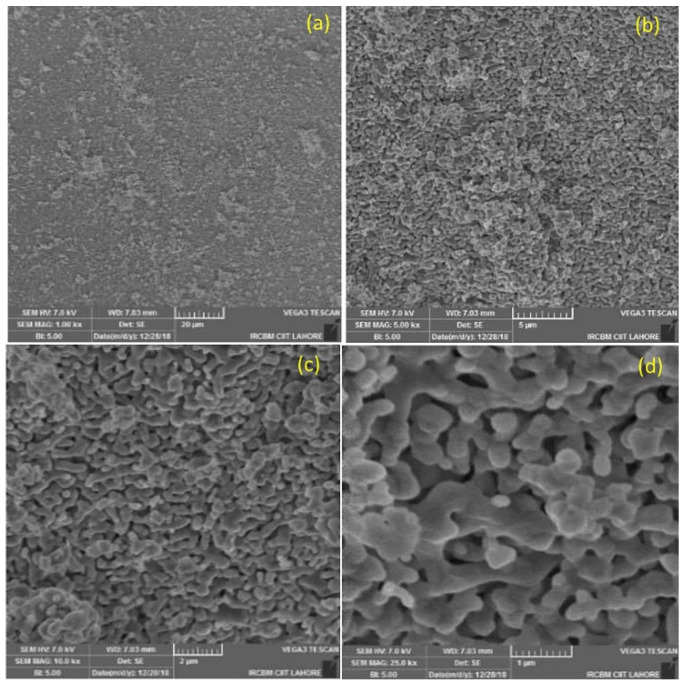
SEM of Si-HA granules at (**a**) 1000×, (**b**) 5000×, (**c**), 10,000×, and (**d**) 25,000×.

**Figure 4 materials-15-01826-f004:**
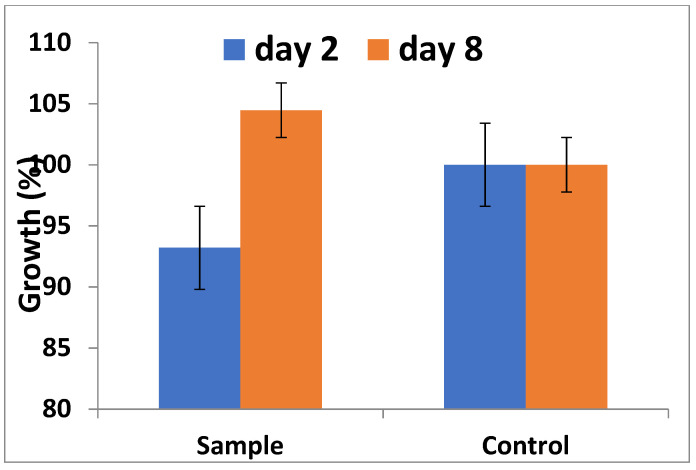
Graphical representation of Alamar Blue assay showing cell proliferation after two days and eight days after of incubation.

**Figure 5 materials-15-01826-f005:**
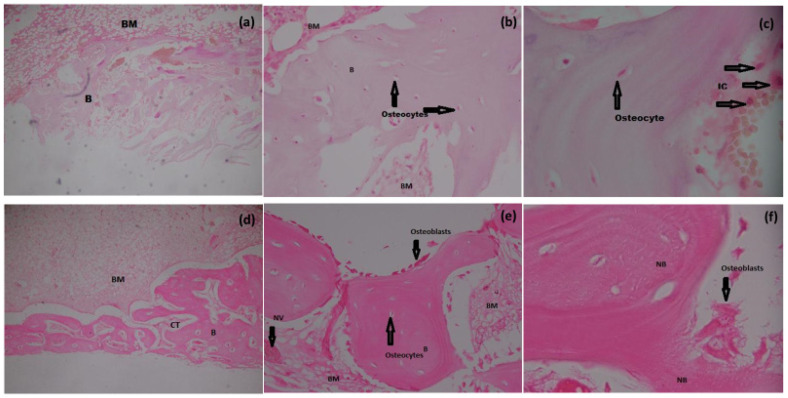
Histological analysis of bone formation at six weeks in Group I (**a**) 10×, (**b**) 40×, (**c**) 100× oil immersion) and Group II (**d**) 10×, (**e**) 40×, (**f**) 100× oil immersion). H&E staining showing new bone formation, osteocytes, osteoblasts, new vessel formation, and inflammatory cells (arrows). Key: B; Bone, BM; Bone marrow, CT; Connective tissue, IC; Inflammatory cells, NB; New bone. Bar = 500 µm.

**Figure 6 materials-15-01826-f006:**
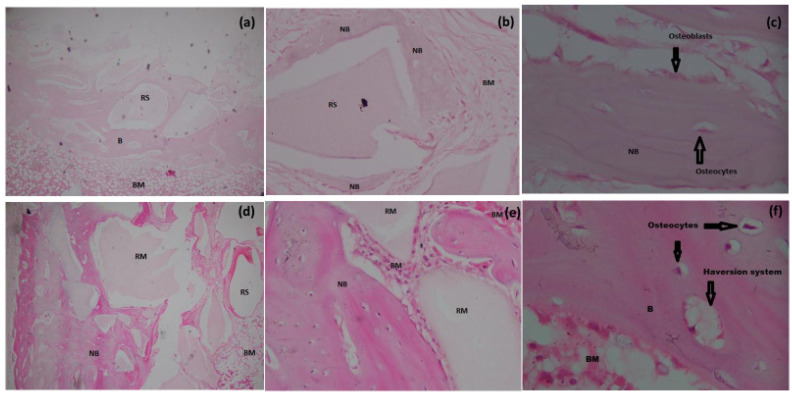
Histological analysis of bone formation at six weeks in Group III (**a**) 10×, (**b**) 40×, (**c**) 100× oil immersion) and Group IV (**d**) 10×, (**e**) 40×, (**f**) 100× oil immersion). H&E staining showing new bone formation, osteocytes, osteoblasts, new vessel formation, and Haversion system (arrows). Key: B; Bone, BM; Bone marrow, CT; Connective tissue, NB; New bone, RM; residual material, RS; Residual spaces. Bar = 500 µm.

**Figure 7 materials-15-01826-f007:**
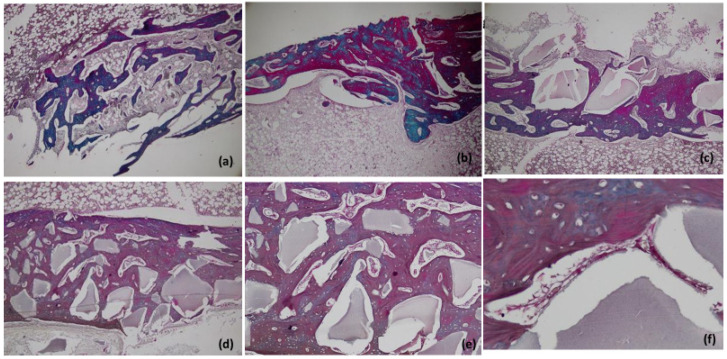
Histological analysis with Masson Trichrome staining of collagen deposition at six weeks, at 5× in Group I (**a**), Group II (**b**), Group III (**c**), and Group IV (**d**). Same at 40× (**e**) and 100× oil immersion (**f**). Bar = 500 µm.

**Figure 8 materials-15-01826-f008:**
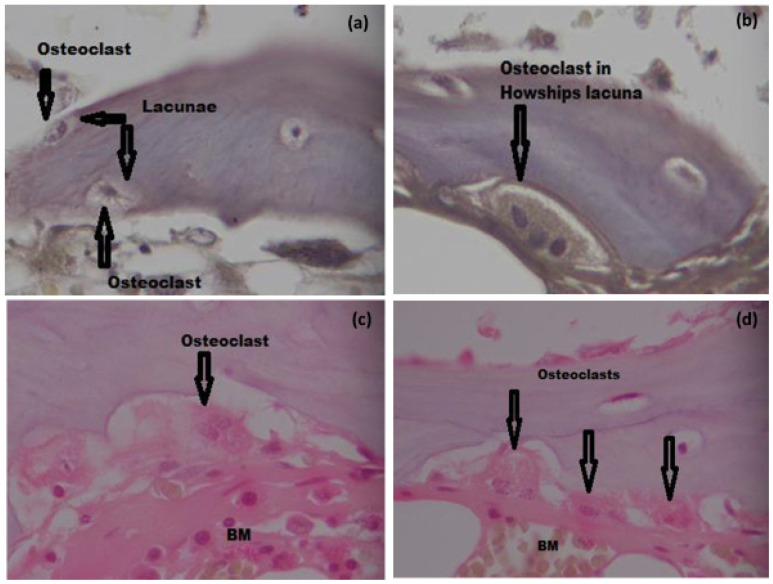
Histological analysis for Osteoclasts (arrows). TRAP staining: (**a**) (40×), (**b**) (100×). H&E staining: (**c**,**d**) (100× oil immersion). Key: BM; Bone marrow Bar = 500 µm.

**Figure 9 materials-15-01826-f009:**
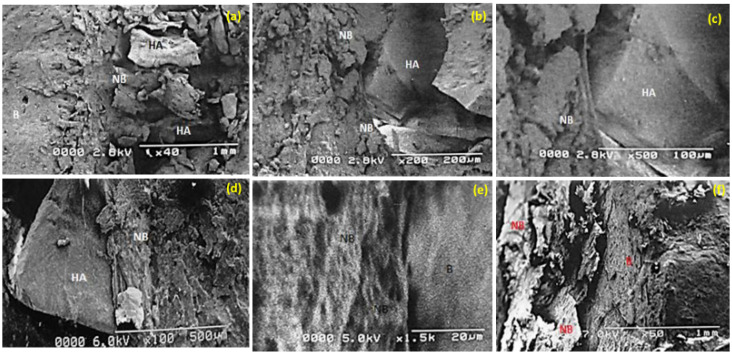
SEM analysis of Si-HA bone interface at six weeks in group IV: (**a**) (2.8 kv, 40×, resolution 1 mm) and (**d**) (6 kv, 100×, 500 µm), group III; (**b**) (2.8 kv, 200×, 200 µm) and (**c**) (2.8 kv, 500×, 100 µm), group II; (**e**) (5 kv, 1.5×, 20 µm) and (**f**) (7 kv, 50×, 1 mm). Key: NB; newly formed bone, B; bone at the defect margin, HA; implanted Si-HA.

**Table 1 materials-15-01826-t001:** Grouping (n = 5) and relevant details of study samples (rabbits).

S. No	Groups		Sample Size	Bone Reconstruction Procedure
1.	A	Control Group	5	Defect was kept as such
2.	B	Experimental group 1	5	Defect was closed by placing bioceramic alone
3.	C	Experimental group 2	5	Defect was closed by placing bioceramic along with SVF * (enzymatic dissociation)
4.	D	Experimental group 3	5	Defect was closed by placing bioceramic along with SVF * (non-enzymatic, modified)

* SVF = Stromal vascular fraction.

**Table 2 materials-15-01826-t002:** Histological evaluation as per histological healing evaluation performance.

S. No	Perimeter	Grading				
		1	2	3	4	5
1	Bone formation	Absent	Present at the periphery	Centrally	Present centrally & at the periphery	-
2	Haversion canals					
3	Osteoblasts					
4	Osteocytes					
5	Osteoclasts					
6	Mature bone					
7	Immature bone					
8	Bone formation closing the defect	Absent	Present between 1% to 25%	Present between 26% to 50%	Present between 51% to 75%	Present between 76% to 100%
9	Vascularization	Absent	Present at the surface	Present in depth	-	-
10	Inflammation	Present	Absent	-	-	-
11	Granulation tissue	Present	Absent	-	-	-

**Table 3 materials-15-01826-t003:** Histological scoring system (severity index).

S. No	Categories		Scores			
			None	Mild	Moderate	Severe
1.	Medullary canal	Leukocytes	0	1	2	3
		Micro abscesses	0	1	2	3
		Fibrosis	0	1	2	3
		Granulation tissue	0	1	2	3
2.	Cortex	Destruction of cortex	0	1	2	3
		Enlarged Haversian canals	0	1	2	3
		Leukocytes	0	1	2	3
		Micro abscesses	0	1	2	3
		Granulation tissue	0	1	2	3
3.	New bone formation		(>50%) 0	(25–50%) 1	(1–25%) 2	(0%) 3
4.	Maximum (worst) score	30				

**Table 4 materials-15-01826-t004:** Results of hemolysis assay.

S. No	Sample No	OD Sample	OD −ve Control	OD +ve Control	% Hemolysis
1	Sample 1	0.079	0.04	1.82	2.19
2	Sample 2	0.018	0.04	1.82	1.23
3	Sample 3	0.102	0.04	1.82	3.66

**Table 5 materials-15-01826-t005:** Mean and standard deviation of histological healing evaluation grading score between groups (ANOVA).

Parameters	Gp-1 (n = 5)	Gp-2 (n = 5)	Gp-3 (n = 5)	Gp-4 (n = 5)	*p*-Value
Histological Evaluation Grading Score	10.20 ± 0.44	26.40 ± 1.34	27.20 ± 2.16	25.00 ± 1.41	<0.001

**Table 6 materials-15-01826-t006:** Inter-group comparison of histological evaluation grading score (post hoc analysis).

Group Comparison	Gp-1 vs. Gp-2	Gp-2 vs. Gp-3	Gp-1 vs. Gp-3	Gp-2 vs. Gp-4	Gp-1 vs. Gp-4	Gp-3 vs. Gp-4
Histological Evaluation Grading Score	<0.001 (*p* value)	0.826 (*p* value)	<0.001 (*p* value)	0.460 (*p* value)	<0.001 (*p* value)	0.126 (*p* value)

**Table 7 materials-15-01826-t007:** Histological grading score (severity index).

S. No	Animal No	Group I	Group II	Group III	Group IV	Group V
1	1	2	2	2	2	1
2	2	2	2	2	1	2
3	3	2	3	2	1	1
4	4	2	3	2	2	1
5	5	2	2	3	2	1
	**Sum**	10	12	11	8	6
	**Mean**	2	2.4	2.2	1.6	1.2

## Data Availability

The data will be made available on request.

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
