# Peer review of "Osteogenic Induction with Silicon Hydroxyapatite Using Modified Autologous Adipose Tissue-Derived Stromal Vascular Fraction: In Vitro and Qualitative Histomorphometric Analysis"

_materials, 2022, doi:10.3390/ma15051826_

Round 1

Reviewer 1 Report

Article is very interesting and could be of interest for the readers. Before publication I would suggest some minor improvements:

Introduction:

The authors also mention in a sentence the use of stem cells in bone regenerative procedures in all fileds, for example dentistry. Please cite: doi: 10.2174/1566523220999200818115803. 

Please add hypothesis at the end of introduction and accept or refuse it at the beginning of discussion.

Could you support the following sentence with a reference?

"The regenerative ability of SVF depends not only on ASCs but also released soluble factors. Such as paracrine factors VEGF and ET-1 both which could promote differentia- tion of osteoblasts. "

Is the severity index referenced somehow in the literature? Could you add a reference?

"The severity index was Score 0 (absence of infection/in- flammation) to Score 3 (severe infection/inflammatory reaction) (Table 3). Stained sections were analyzed and photographed with microscope (Olympus-CX41, Olympus, Japan) connected to a CCD camera (DP72; Olympus, Japan) using cell Sens standard software (Olympus, Japan) [39]. "

Please add the limitations of this study.

Reviewer 2 Report

This work, of course, is of certain interest for people interested in  biocompatibility of Silicon hydroxyapatite (Si-HA), but in its current form cannot be recommended for publication.

  1. The introduction does not reflect the scope of the journal Materials, Furthermore, the motivation for this new study, as well as the existing background of hydroxyapatite research, are very poorly reflected.
  2. Most of the references are outdated (over 10 years old) and it is unclear if this is still an interesting topic and what has been done in this direction in recent years.
  3. Important question is question about sample aging during time as well during radiation treatment. As we know from literature, apatite materials are subjected to ionizing and particle irradiation. See, for example, Bystrova, A.; Dekhtyar, Y.D.; Popov, A.; Coutinho, J.; Bystrov, V. Modified hydroxyapatite structure and properties: Modeling and synchrotron data analysis of modified hydroxyapatite structure. Ferroelectrics2015475, 135–147.

Hübner, W.; Blume, A.; Pushnjakova, R.; Dekhtyar, Y.; Hein, H.-J. The influence of X-ray radiation on the mineral/organic matrix interaction of bone tissue: An FT-IR microscopic investigation. Int. J. Artif. Organs 200528, 66–73.

  1. Paragraph 2.1. Here it is not at all clear how the obtained samples can be compared with others used in the world, including commercial ones.
  2. Can the authors convince us that the use of two methods (XRD and SEM) is sufficient to characterize samples?
  3. Furthermore, data on Fig.1 needs additional discussion, because good match of XRD patterns looks unconvincing.
  4. Before conclusions, the authors should clearly articulate what new data for the scientific material research community were obtained.
  5. Unfortunately, not all references are available, therefore, where the authors cannot give doi, it is recommended to replace these references with those that will be available to readers.

Round 2

Reviewer 2 Report

It looks that the article has been sufficiently improved and can now be recommended for publication.